# Risk factors for osteoporosis in elderly patients with type 2 diabetes: A protocol for systematic review and meta-analysis

**Wenhao Su, Hairong Jia, Luo Yang, Jiaqi Zhang, Zhaoyang Wei, Pepertual Tsikwa, Yanru Wang** [ORCID]*

School of Nursing, Zhejiang Chinese Medical University, Hangzhou, Zhejiang, China

* Wangyanru001@outlook.com

## Abstract

### Background

Osteoporosis is a prevalent chronic result of diabetes. Osteoporosis susceptibility is raised by unstable blood glucose levels, oxidative stress, hormonal abnormalities, and other factors. Currently, there is no systematic review addressing the risk factors of osteoporosis in diabetes. This study intends to systematically assess the current risk factors related to diabetic osteoporosis (DOP) and provide suggestions for the improvement of therapy approaches.

### Methods and analysis

We will search five English literature databases (PubMed, Embase, Web of Science, CINAHL, and Cochrane Library) and three Chinese databases (CNKI, WanFang, and SinoMed) from the starting point until December 31, 2024. We will perform a systematic examination and meta-analysis of cohort and case-control studies to identify all population-based risk factors for diabetic osteoporosis. Two researchers will independently assess the publication, extract data, and evaluate the quality and potential biases present in the study. We will utilize RevMan V.5.4 software and STATA 16.0 for data analysis. The included studies will be assessed using the Newcastle Ottawa Quality Assessment Instrument (NOS). If the heterogeneity of the included studies is higher than 50%, we will perform subgroup and sensitivity analysis to identify probable sources of heterogeneity. The assessment of publication bias will be conducted using funnel plot. Furthermore, we will employ the Grading of Recommendations Assessment, Development, and Evaluation (GRADE) to assess the quality of evidence for each exposure and outcome.

### Discussion

This protocol aims to investigate the risk variables associated with DOP. We will summarize the current knowledge about factors influencing osteoporosis in diabetes. We strive to assist physicians with more extensive references for decision-making and facilitate the implementation of effective prevention strategies for DOP.

**Data availability statement:** No datasets were generated or analysed during the current study. All relevant data from this study will be made available upon study completion.

**Funding:** The author(s) received no specific funding for this work.

**Competing interests:** The authors have declared that no competing interests exist.

**Abbreviation:** DOP, Diabetic Osteoporosis; NOS, The Newcastle Ottawa Quality Assessment Instrument; GRADE, Grading of Recommendations, Assessment, Development, and Evaluation; APCO, The Asia Pacific Osteoporosis Association; BMD, The Bone Mineral Density; SGT2I, Sodium-Glucose Cotransporter-2 Inhibitors; PRISMA-P, Preferred Reporting Items for Systematic reviews and Meta-Analyses; MOOSE, Meta Analysis of Epidemiological Observational Studies; BMI, Body Mass Index; OR, Odd Ratio; CI, Confidence Interval; SMD, Standard Mean Deviation; WMD, Weighted Mean Deviation

## Registration

This study has been registered in the PROSPERO (CRD42024602637).

## Introduction

Diabetes is a prevalent chronic non-communicable disease, which has become a serious public health problem in the world. Around 537 million persons are impacted by diabetes [1]. By 2030, this figure is anticipated to rise to 643 million. Diabetes has resulted in economic losses of no less than 966 billion US dollars, a fourfold increase since 2009[2]. The prolonged abnormal blood glucose levels in diabetes can lead to microvascular complications (such as diabetic retinopathy and diabetic nephropathy), macrovascular complications (such as coronary artery disease, heart failure, and peripheral arterial disease), neuropathy, and diabetic foot, as well as disrupt bone metabolism and induce osteoporosis [3].

The incidence of osteoporosis in diabetes patients is increasing alongside the progression of diabetes [4]. Osteoporosis is a chronic metabolic bone disorder marked by reduced bone mass and microstructural deterioration of bone tissue, resulting in heightened bone fragility and an elevated risk of fractures [5]. Abnormal bone metabolism in diabetic patients can diminish bone density and compromise bone structure [6], with osteoporosis prevalence reaching 27.67% [6], The risk of osteoporosis in type 2 diabetes patients is significantly higher than that of ordinary people. Research shows that the risk of fracture healing damage in diabetes patients is 2.11 times that in non-diabetes patients [7]. Diabetes increases the fracture risk by destroying bone metabolism and accumulating advanced glycation end products (AGEs), further exacerbating bone fragility [8]. Moreover, the risk of osteoporosis increases with the prolongation of the course of diabetes [9] and leads to fractures, falls, brain injuries, and other adverse consequences [10].

In addition, the risk of osteoporosis in elderly diabetes patients is significantly higher than other diabetes patients. On the one hand, the physical function of elderly people degenerates with age, and calcium content in bones decreases, resulting in osteoporosis; On the other hand, elderly patients with diabetes are more likely to suffer from sleep disorders, which are closely related to low bone density and osteoporosis [11]. Sleep disorders can inhibit the function of osteoblasts, promote bone absorption and decomposition, weaken the ability of bone microstructure damage repair accumulated with age, and lead to a higher risk of osteoporosis [12]. Long-term diabetes impairs renal function, resulting in diminished synthesis of vitamin D, which subsequently affects intestinal calcium absorption [13]. Insulin regulates osteoblasts; however, in diabetic individuals, inadequate insulin results in impaired osteogenesis and insufficient bone production [14]. All the aforementioned factors cause individuals with diabetes more susceptible to osteoporosis.

Though diabetic osteoporosis carries serious effects, it has not attracted much attention. The Asia Pacific Osteoporosis Association (APCO) indicates that merely 31% of diabetic patients have undergone bone health evaluations [3]. Moreover, the bone mineral density (BMD) of individuals with type 2 diabetes is greater than that of age-matched non-diabetic individuals [9,15], however, BMD does not correlate directly with bone strength. Consequently, individuals may undervalue the risk of osteoporosis in diabetic patients.

Several hypoglycemia therapeutic interventions could influence bone metabolism [16], and certain diabetic medications may expedite the progression of osteoporosis. Thiazolidinedione (TZD) could elevate the risk of weight gain, edema, fractures, and heart failure in diabetic patients [17]. Sodium-glucose cotransporter-2 inhibitors (SGT2I) may influence bone resorption by altering electrolyte homeostasis, calcium and phosphorus metabolism, while also

elevating parathyroid hormone levels and diminishing 1,25-(OH)-vitamin D concentrations [18]. Because diabetes is a long-term disease that affects the whole body, it usually takes longer for people with diabetes to heal from fractures. This makes them more likely to get hospital-acquired infections, muscle loss, and pneumonia [19].

The accumulation of the receptor of advanced glycation endproducts (RAGE) in the cortical bone of diabetes patients will harden type I collagen in the bone matrix, reduce bone strength, increase bone fragility, and promote osteoblast apoptosis [20]. In addition, the course of diabetes [21], insulin resistance [22], bone marrow fat accumulation, and low bone turnover level are all factors that cause the increased risk of fracture in diabetes patients [20]. Therefore, people may underestimate the risk of osteoporosis in patients with diabetes.

Although it is known that abnormal blood glucose in diabetes can cause osteoporosis, the pathogenesis and influencing factors of osteoporosis in type 2 diabetes are still unclear. There are few studies on osteoporosis in elderly patients with type 2 diabetes, and the risk factors pointed out in various studies are different [23–27]. Therefore, the purpose of this study is to identify the risk factors of DOP through meta-analysis, to help medical staff identify high-risk groups in time and take effective intervention strategies.

## Methods

### Study registration

This research protocol has been registered with PROSPERO (CRD42024602637). We will follow the Preferred Reporting Items for Systematic Reviews and Meta-Analyses (PRISMA-P) guidelines [28] (S1 File) and report by the PRISMA Statement [29] and the Meta-Analysis of Epidemiological Observational Studies (MOOSE) guidelines [30](S2 File).

### Ethics and dissemination

This study doesn't require patient informed consent or approval from the ethics committee because it is a meta-analysis protocol, not a clinical trial. The findings of the systematic review and meta-analysis will be shared in peer-reviewed journals.

### Eligibility Criteria

**Participants.** We will incorporate elderly people diagnosed with osteoporosis and diabetes. Individuals aged 60 years or older are classified as elderly [31]. Diabetic osteoporosis can be characterized by a physician's diagnosis (either confirmed or self-reported by the patient) or by the author's assertion that the research subject pertains to the diabetic osteoporosis population.

**Exposure.** The primary outcome measures will be patient characteristics that may serve as risk factors or predictors of worsening. This may encompass, but is not restricted to, demographic attributes (including age, gender, race/ethnicity), osteoporosis-related characteristics in patients (such as disease duration, medication, and bone density), and additional health-related factors (such as smoking, body mass index (BMI), comorbidities, and concomitant medications).

**Types of studies.** Only case-control studies and cohort studies will be considered

### Exclusion criteria

We will reject case-control studies and cohort studies involving people whose primary health concern is not DOP, as well as studies without comprehensive data retention. Studies will be rejected if they fulfill the following criteria:

(1) Repeated publications, sessions, meta-analyses, reviews, protocols, animal experiments, and letters;

(2) Inaccessibility of comprehensive literature or insufficient existing data;

(3) Low-quality studies. The Newcastle Ottawa Quality Assessment Tool (NOS) suggests low quality in the literature (NOS score < 4).

## Search strategy

We will examine the following databases: PubMed, Web of Science, CINAHL, Cochrane Library, EMBASE, CNKI, WanFang, and SinoMed. Furthermore, we will seek grey literature and manually obtain the references cited in the paper to ensure no relevant research is overlooked. This study will utilize medical subject headings (MeSH) and keywords for the search, encompassing the time from the foundation of the database to December 31, 2024. Comprehensive details regarding the search strategy are available in the attached file (S3 File). The search terms contain diabetes, osteoporosis, and associated risk factors.

## Data collection and analysis

We will import all obtained studies into Endnote X9 software and eliminate all duplicate studies. Then, the remaining articles will be uploaded to the Rayyan website for independent assessment of titles, abstracts, and texts by two trained researchers (W.H. and H.R.) and two independent researchers (Z.Y. and J.Q.) respectively to eliminate studies that did not satisfy the inclusion criteria. If the researchers fail to achieve consensus on the aforementioned two steps, the ultimate decision will be rendered by the third researcher (Y.R. or L.Y.). The research selection process is shown in S1 Fig.

## Data extraction

Two researchers (W.H. and Z.Y.) independently screen literature and conduct cross-checking. When there is a disagreement, the final decision will be made by the third researcher (Y.R.). Extract data using a pre-designed data table by the researcher, which mainly includes the following information: first author, publication year, study type, study location, country, sample size, gender, age, treatment medication, medication adjustments, osteoporosis-related influencing factors, OR values and 95% CI of risk factors, etc. If information is missing, we will contact the first author or corresponding author every Monday and Friday for two consecutive months. If no response is received, the study will be included in the research, but only a narrative description will be provided.

## Assessment of risk of bias

The selected articles will be assessed by two qualified researchers (H.R. and J.Q.) utilizing the Newcastle Ottawa Scale (NOS) (S4 File). In the event of a disagreement, the third researcher (Y.R. or L.Y.) will be consulted to make the final decision. The quality rating score of each study will be listed in the study's basic information table. The studies assessed utilized the NOS, comprising eight variables including population selection, inter-group comparability, and outcome measurement. All items are assigned 1 point, except for inter-group comparability, which is assigned 2 points. The NOS has a maximum score of 9 points, with scores of 7 or above indicating high quality, scores between 5 and 6 suggesting medium quality, and scores of 4 or below indicating bad quality. This study will exclusively incorporate high-quality and medium-quality literature, whereas low-quality literature will be omitted. We will employ

GRADE (Grading of Recommendations, Assessment, Development, and Evaluation) to assess the accuracy of the meta-analysis findings.

## Strategy for data synthesis

Our study will utilize RevMan V.5.4 software to do a meta-analysis of risk factors identified in the literature collected. We will utilize Stata 16.0 to amalgamate data on influencing factors from three or more studies. The categorical variables are denoted by odds ratio (OR) and 95% confidence interval (CI), with $P < 0.05$ signifying statistically significant differences. The binary data will be examined utilizing the standard mean deviation (SMD) or weighted mean deviation (WMD) with a 95% CI. In heterogeneity tests, $I^2 < 50\%$ and $P > 0.05$ indicate minimal heterogeneity, warranting the adoption of fixed effects models for analysis; conversely, $I^2 \geq 50\%$ and $P \leq 0.05$ signify the presence of heterogeneity. If heterogeneity remains $\geq 50\%$ after eliminating evident sources of clinical heterogeneity, a random effects model analysis will be employed. Substantial heterogeneity will be examined utilizing Stata 16.0 software, focusing on age, gender, geography, sample size, medication (dosage, type of medication), treatment duration, and various risk factors via subgroup or sensitivity analysis. If heterogeneity exceeds 75%, a meta-analysis will not be performed. We will employ descriptive analysis.

## Quality of evidence and publication biases assessment

Two researchers (W.H. and H.R.) will use GRADE to evaluate the quality of evidence. When the literature on any risk factor in the study is > 10, we will employ Egger's test to assess the presence of publication bias. Obvious asymmetry on both sides of the funnel plot suggests the potential for publishing bias. A p-value below 0.05 suggests the presence of publication bias. Furthermore, if p > 0.05, it indicates a reverse funnel plot outcome, and we will employ the cut-and-patch method for further analysis.

## Sensitivity analysis

In the presence of substantial heterogeneity in the included studies, indicated by an $I^2$ value exceeding 75%, we will conduct subgroup and sensitivity analyses to investigate the sources of heterogeneity further. We will employ funnel plots and the Egger test to assess publication bias.

## Discussion

Currently, there is no comprehensive assessment to thoroughly analyze the risk factors of osteoporosis diabetic patients. This study will be the first comprehensive investigation into the risk factors for DOP. This review's findings may enhance health outcomes for older diabetes patients by facilitating early identification and strengthening the surveillance or management of diabetic osteoporosis, hence mitigating deteriorating risk factors.

   Diabetes is a chronic endocrine metabolic disorder, and prolonged hyperglycemia [32] and oxidative stress [33] will lead to osteoporosis in diabetic patients. Secondly, with advancing age, variables such as diminished sex hormone levels [34] and insufficient physical activity elevate the risk of osteoporosis in elders. Factors including diminished digestive function, impaired movement, and reduced vision in the elderly exacerbate their vulnerability to fractures following osteoporosis [35]. Moreover, individuals with diabetes frequently exhibit obesity and hyperlipidemia, and data suggests that this correlates with elevated bone density and diminished bone strength [36,37]. Although the bone mineral density of diabetes patients is normal or higher, the fracture risk of diabetes patients increases by 40–70% [15,38].

Consequently, diabetic patients tend to get fractures despite elevated bone density levels [15]. The chronic complications of diabetes, including diabetic nephropathy [39], diabetic liver disease [40], and diabetic retinopathy [41], along with the coexistence of multiple chronic conditions such as hypertension, coronary heart disease, and heart failure, hinder the clinical management of elderly diabetic patients and may elevate their mortality risk [42]. Some studies indicate that the risk of hip fracture in diabetic patients is 1.7 times more than that of the general population, and the mortality rate following hip fracture is similarly elevated compared to non-diabetic individuals [43]. Many studies have investigated diverse risk factors associated with osteoporosis in patients with diabetes. The determinants encompass age, gender [44,45], pharmacological agents (such as thiazolidinediones) [46], comorbidities (including diabetic nephropathy [47], hypertension [48]), and the disease's severity. Certain medications not only effectively regulate blood glucose levels but also contribute positively to the prevention of osteoporosis, including metformin [49] and dipeptidyl peptidase-4 inhibitors [50]. The enduring protective efficacy of the aforementioned medications on bone health requires more validation. Upon meticulous analysis of the available data, we detected deficiencies in prior research. Numerous research primarily investigates the risk factors, prevention, and treatment of osteoporosis in the aged population. There exists a distinction between the osteoporosis observed in senior diabetic patients and that in the general elderly population. The habitual application of X-ray for assessing the risk of osteoporosis does not reliably indicate the risk of osteoporosis or fractures in older individuals with diabetes. Diabetes significantly affects bone density, bone matrix, bone structure, and bone metabolism [51]. X-rays alone are insufficient for making precise assessments. Consequently, it is imperative to conduct a thorough evaluation of the risk factors for osteoporosis in older individuals with diabetes to identify all pertinent risk variables and establish appropriate preventative or treatment measures.

## Limitation

This study may have some potential limitations. (1) We plan to only include observational studies, which may not cover all the influencing factors of DOP and can only obtain modifiable risk and protective factors that exist in the existing evidence base. (2) This study will be limited to Chinese and English studies and may overlook important research in other languages. However, considering that team members are proficient in both Chinese and English, using translation software to translate other languages may result in translation errors, leading to significant deviations in the results. (3) This article only focuses on osteoporosis in elderly patients with diabetes. The osteoporosis situation of young diabetes patients and postmenopausal female diabetes patients is also not optimistic. In the future, research can be carried out on this kind of population, and the common influencing factors of osteoporosis in young and old diabetes patients may be found. (4) Since we will include all the studies on risk factors of osteoporosis in diabetes patients, the number of literature on some influencing factors may be small, so we will use the form of table and description to elaborate, and the research results may also have high heterogeneity, so we will use subgroup analysis and sensitivity analysis to interpret the heterogeneity, and use narrative description to interpret some results. (5) To include as much literature as possible for analysis, we will also search for grey literature. However, grey literature may have lower quality and affect the outcome, so we will only include grey literature with higher research quality.

## Conclusion

This study will strictly follow various guidelines (including PRISMA, NOS, MOOSE, GRADE) to ensure that our research complies with reporting standards. Searching 8 databases in both

Chinese and English can obtain DOP epidemiological data from different cultural backgrounds. This study will explore the heterogeneity found under different age, gender, region, sample size, treatment drugs (drug dosage, drug type), treatment time, and different types of influencing factors. In addition, identifying the risk and protective factors of DOP will enhance our understanding of DOP patients. This study aims to explore the modifiable risk factors and potential protective factors in DOP patients, which is of great significance for clinical practice. The results of the study may help to develop a health promotion and disease prevention plan for type 2 diabetes aimed at reducing DOP.

## Supporting information

**S1 Fig. PRISMA flow diagram of the selection process.**
(TIF)

**S1 File. PRISMA-P.**
(PDF)

**S2 File. MOOSE (Meta-analyses Of Observational Studies in Epidemiology) Checklist.**
(PDF)

**S3 File. Search strategy.**
(DOCX)

**S4 File. The Newcastle Ottawa quality assessment instrument (NOS).**
(DOC)

## Author contributions

**Conceptualization:** Wenhao Su.

**Data curation:** Jiaqi Zhang.

**Formal analysis:** Wenhao Su, Hairong Jia, Zhaoyang Wei, Yanru Wang.

**Investigation:** Jiaqi Zhang, Zhaoyang Wei.

**Methodology:** Hairong Jia, Luo Yang, Yanru Wang.

**Supervision:** Luo Yang, Yanru Wang.

**Writing – original draft:** Wenhao Su, Hairong Jia, Pepertual Tsikwa.

**Writing – review & editing:** Wenhao Su, Luo Yang, Yanru Wang.

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
