## [Decision Letter · Decision Letter 0]

8 Dec 2024

PONE-D-24-50961Risk factors for osteoporosis in elderly patients with type 2 diabetes: a protocol for systematic review and meta-analysisPLOS ONE

Dear Dr. Yanru,

Thank you for submitting your manuscript to PLOS ONE. After careful consideration, we feel that it has merit but does not fully meet PLOS ONE’s publication criteria as it currently stands. Therefore, we invite you to submit a revised version of the manuscript that addresses the points raised during the review process.

We look forward to receiving your revised manuscript.

Kind regards,

Amirhossein Ghaseminejad-Raeini

Academic Editor

PLOS ONE

Reviewers' comments:

Reviewer's Responses to Questions

**Comments to the Author**

1. Does the manuscript provide a valid rationale for the proposed study, with clearly identified and justified research questions?

Reviewer #1: Yes

Reviewer #2: Partly

2. Is the protocol technically sound and planned in a manner that will lead to a meaningful outcome and allow testing the stated hypotheses?

Reviewer #1: Yes

Reviewer #2: Yes

3. Is the methodology feasible and described in sufficient detail to allow the work to be replicable?

Reviewer #1: No

Reviewer #2: Yes

4. Have the authors described where all data underlying the findings will be made available when the study is complete?

Reviewer #1: Yes

Reviewer #2: Yes

5. Is the manuscript presented in an intelligible fashion and written in standard English?

Reviewer #1: Yes

Reviewer #2: No

6. Review Comments to the Author

You may also provide optional suggestions and comments to authors that they might find helpful in planning their study.

Reviewer #1: Thank you for inviting me to review this study protocol. The topic is interesting and the protocol is well-written.

I believe that applying the following comments will improve the protocol:

- line 112: it is suggested to add your reference for this classification.

- it is suggested to include data regarding the medications of patients and adjustments in the statistical analysis as well.

-Data extraction part is somehow general and needs to be more consice.

Reviewer #2: The specific questions and recommendations are as follows:

1:Although the Introduction section offers a suitable framework, it is lacking in details and would benefit

from being expanded. It is advised to include more references in this area. The primary text's conclusion

also needs to be extended and revised. Think about including additional evidence.

2:After the Discussion section, I suggest adding a new section regarding limitations and adding additional

details about them, along with suggestions for how to resolve these problems in current and future

research. A more thorough description would be helpful.

3:the references.

It is advised that a new section explaining any acronyms used in the article be added at the end, before

4:There is no Conclusion in the manuscript. The authors are advised to create a dedicated section for concluding

points at the end of the article.

5:The discussion should be more comprehensive. It should be more vigorously approached and a detailed explanation

into the issue is required. Adding more references is suggested.

6:The manuscript is generally clear and concise; however, minor grammatical adjustments could enhance readability.

Thanks to the authors for their hard work on this Study. It’s a pleasure to review such a contribution to the

field.

7. PLOS authors have the option to publish the peer review history of their article (what does this mean? ). If published, this will include your full peer review and any attached files.

**Do you want your identity to be public for this peer review?** For information about this choice, including consent withdrawal, please see our Privacy Policy .

Reviewer #1: No

Reviewer #2: **Yes: ** SEYED AMIRHOSSEIN MAZHARI

---

## [Author Response · Author response to Decision Letter 1]

27 Dec 2024

Greatly appreciate reviewers' comment, thank you for your kind advice.

---

## [Decision Letter · Decision Letter 1]

24 Jan 2025

PONE-D-24-50961R1Risk factors for osteoporosis in elderly patients with type 2 diabetes: a protocol for systematic review and meta-analysisPLOS ONE

Dear Dr. Wang,

Thank you for submitting your manuscript to PLOS ONE. After careful consideration, we feel that it has merit but does not fully meet PLOS ONE’s publication criteria as it currently stands. Therefore, we invite you to submit a revised version of the manuscript that addresses the points raised during the review process.

We look forward to receiving your revised manuscript.

Kind regards,

Amirhossein Ghaseminejad-Raeini

Academic Editor

PLOS ONE

Journal Requirements:

Reviewers' comments:

Reviewer's Responses to Questions

**Comments to the Author**

1. Does the manuscript provide a valid rationale for the proposed study, with clearly identified and justified research questions?

Reviewer #1: Partly

Reviewer #2: Yes

2. Is the protocol technically sound and planned in a manner that will lead to a meaningful outcome and allow testing the stated hypotheses?

Reviewer #1: Partly

Reviewer #2: Yes

3. Is the methodology feasible and described in sufficient detail to allow the work to be replicable?

Reviewer #1: Yes

Reviewer #2: Yes

4. Have the authors described where all data underlying the findings will be made available when the study is complete?

Reviewer #1: No

Reviewer #2: Yes

5. Is the manuscript presented in an intelligible fashion and written in standard English?

Reviewer #1: Yes

Reviewer #2: Yes

6. Review Comments to the Author

You may also provide optional suggestions and comments to authors that they might find helpful in planning their study.

Reviewer #1: The authors have addressed my comments, completely. I have no further comments for them. I only suggesst to make all data available as there is no restrictions for using data from already published papers.

Reviewer #2: The manuscript is analyzed appropriately and provides suitable results. Additionally, adherence

to multiple guidelines, ensures transparency and accuracy in the methodology. The Discussion

section needed to be more vigorously approached and a detailed explanation into the issue is

required. Fortunately, after extensive revision the discussion part is now more all-inclusive,

additionally limitation are better presented in the revised version. The overall language has

improved as well.

At the end I have some questions about this article.

The specific questions and recommendations are as follows:

1:

Although the Introduction section offers a suitable framework, it is lacking in details and would benefit

from being expanded. It is advised to include more references in this area. The primary text's conclusion

also needs to be extended and revised. Think about including additional evidence.

Appropriately corrected, both introduction and conclusion are now informative.

2:

After the Discussion section, I suggest adding a new section regarding limitations and adding additional

details about them, along with suggestions for how to resolve these problems in current and future

research. A more thorough description would be helpful.

A new section has been created and limitations are explained in the dedicated section.

3:

It is advised that a new section explaining any acronyms used in the article be added at the end, before the references.

New section has been incorporated into the text regarding any acronym used in the main text.

4:

There is no Conclusion in the manuscript. The authors are advised to create a dedicated section for concluding

points at the end of the article.

The structure of the text is now more standardized and conclusions are being discussed at the end of the manuscript.

5:

The discussion should be more comprehensive. It should be more vigorously approached and a detailed explanation

into the issue is required. Adding more references is suggested.

Appropriately addressed.

6:

The manuscript is generally clear and concise; however, minor grammatical adjustments could enhance readability.

Overall language and readability are improved and the text grammar has been enhanced.

Thanks to the authors for their hard work on this Revision. It’s a pleasure to review such a contribution to the

field.

7. PLOS authors have the option to publish the peer review history of their article (what does this mean? ). If published, this will include your full peer review and any attached files.

**Do you want your identity to be public for this peer review?** For information about this choice, including consent withdrawal, please see our Privacy Policy .

Reviewer #1: No

Reviewer #2: **Yes: ** SEYED AMIRHOSSEIN MAZHARI

---

## [Author Response · Author response to Decision Letter 2]

27 Jan 2025

We really appreaciate reviewers kind and careful advice. Hope you all have a nice day!

---

## [Editor Report · Decision Letter 2]

5 Feb 2025

Risk factors for osteoporosis in elderly patients with type 2 diabetes: a protocol for systematic review and meta-analysis

PONE-D-24-50961R2

Dear Dr. Wang,

We’re pleased to inform you that your manuscript has been judged scientifically suitable for publication and will be formally accepted for publication once it meets all outstanding technical requirements.

Kind regards,

Amirhossein Ghaseminejad-Raeini

Academic Editor

PLOS ONE
---

## [Editor Report · Acceptance letter]

PONE-D-24-50961R2

PLOS ONE

Dear Dr. Wang,

I'm pleased to inform you that your manuscript has been deemed suitable for publication in PLOS ONE. Congratulations! Your manuscript is now being handed over to our production team.

Kind regards,

on behalf of

Dr. Amirhossein Ghaseminejad-Raeini

Academic Editor

PLOS ONE